# Evaluating Translational Methods for Personalized Medicine—A Scoping Review

**DOI:** 10.3390/jpm12071177

**Published:** 2022-07-19

**Authors:** Vibeke Fosse, Emanuela Oldoni, Chiara Gerardi, Rita Banzi, Maddalena Fratelli, Florence Bietrix, Anton Ussi, Antonio L. Andreu, Emmet McCormack

**Affiliations:** 1Centre for Cancer Biomarkers, Department of Clinical Science, University of Bergen, 5021 Bergen, Norway; emmet.mc.cormack@uib.no; 2EATRIS ERIC, European Infrastructure for Translational Medicine, 1081 HZ Amsterdam, The Netherlands; emanuelaoldoni@eatris.eu (E.O.); florencebietrix@eatris.eu (F.B.); antonussi@eatris.eu (A.U.); toniandreu@eatris.eu (A.L.A.); 3Centre for Health Regulatory Policies, Istituto di Ricerche Farmacologiche Mario Negri IRCCS, 20156 Milan, Italy; chiara.gerardi@marionegri.it (C.G.); rita.banzi@marionegri.it (R.B.); 4Department of Biochemistry and Molecular Pharmacology, Istituto di Ricerche Farmacologiche Mario Negri IRCCS, 20156 Milan, Italy; maddalena@marionegri.it; 5Centre for Pharmacy, Department of Clinical Science, The University of Bergen, 5021 Bergen, Norway

**Keywords:** personalized medicine, translational models, preclinical models, stratified treatment selection

## Abstract

The introduction of personalized medicine, through the increasing multi-omics characterization of disease, brings new challenges to disease modeling. The scope of this review was a broad evaluation of the relevance, validity, and predictive value of the current preclinical methodologies applied in stratified medicine approaches. Two case models were chosen: oncology and brain disorders. We conducted a scoping review, following the Joanna Briggs Institute guidelines, and searched PubMed, EMBASE, and relevant databases for reports describing preclinical models applied in personalized medicine approaches. A total of 1292 and 1516 records were identified from the oncology and brain disorders search, respectively. Quantitative and qualitative synthesis was performed on a final total of 63 oncology and 94 brain disorder studies. The complexity of personalized approaches highlights the need for more sophisticated biological systems to assess the integrated mechanisms of response. Despite the progress in developing innovative and complex preclinical model systems, the currently available methods need to be further developed and validated before their potential in personalized medicine endeavors can be realized. More importantly, we identified underlying gaps in preclinical research relating to the relevance of experimental models, quality assessment practices, reporting, regulation, and a gap between preclinical and clinical research. To achieve a broad implementation of predictive translational models in personalized medicine, these fundamental deficits must be addressed.

## 1. Introduction

The emergence of personalized medicine (PM) *(refer to*
Box 1
*for definition),* through phenotype presentation and individual omics characterization, demands preclinical models which can recapitulate clinical features and provide predictive “personalized” data. This has meant a shift in translational research away from the “one size fits all” demonstration of efficacy, towards developing more sophisticated and targeted preclinical models. There are many definitions of translational research; in this review we define it as a scientific process to improve human health via a “bench-to-bedside” approach. Translational methods that integrate the increasing molecular categorization of diseases can inform a stratified medicine approach, and potentially predict treatment responses in subgroups of patients with certain biological signatures.

Box 1Personalized medicine.According to the European Council Conclusion on personalized medicine for patients personalized medicine is “a medical model using characterisation of individuals’ phenotypes and genotypes (e.g., molecular profiling, medical imaging, lifestyle data) for tailoring the right therapeutic strategy for the right person at the right time, and/or to determine the predisposition to disease and/or to deliver timely and targeted prevention [1]”. In the context of the PERMIT project, we applied the following common operational definition of personalized medicine research: a set of comprehensive methods, (methodological, statistical, validation, or technologies) to be applied in the different phases of the development of a personalized approach to treatment, diagnosis, prognosis, or risk prediction. Ideally, robust and reproducible methods should cover all the steps between the generation of the hypothesis (e.g., a given stratum of patients could better respond to a treatment), its validation and preclinical development, and up to the definition of its value in a clinical setting.

Rodent models are by far the most used in vivo models in translational research [2]. In fact, much of our current understanding of mechanisms of disease is based on research in these animals, but there is concern about the limited translational power [3,4,5,6,7]. In recent decades, animal models have been refined through precise genetic modifications, and there are well established genetically engineered mouse models (GEMMs) for many diseases [8,9]. The “mouse avatar” concept, where patient tissue is directly xenografted in vivo and studied, is now an established technique in the field of oncology [10,11]. This model has been applied in co-clinical trials, where mouse experiments are developed in parallel with human clinical trials in order to enable real-time transfer of information [12]. Correspondingly, the emergence of advanced in vitro techniques, such as the development of 3D cell cultures and organoids, and the integration of microfluidics to create organ-on-chip technologies, bring the promise of patient-derived personalized cellular models in the future [13,14,15,16,17,18]. Patient-derived cancer organoids are already being employed for drug-screening applications [19]. However, currently, one of the main challenges with both these patient-derived models is the absence of a functional immune system. In silico modeling, referring to mathematical and computational models of biological systems, is another approach which can complement the personalized approach, aiming to make predictions on drug targets, drug efficacy, and patient responses [20,21].

The promise of PM is that each patient’s treatment can be optimally tailored to their disease. Presumably, models—or a combination of models—that can successfully discriminate between responders and non-responders for a given treatment can provide predictive data prior to therapeutic clinical trials (Figure 1). Most preclinical models have been generated to understand disease mechanisms; for instance, defining a specific phenotype or a biological molecular mechanism. However, if preclinical model systems should have a predictive value as well, meaning that the results obtained with the model predict outcomes in humans and can support the decision to initiate and authorize clinical trials in PM, they must generate reliable data. The advancement of preclinical research in this regard is promising, but most of the methodologies are still in their infancy. There is a need for more robust preclinical resources which can validate biomarkers and demonstrate the clinical utility of the stratified approach. The identification of bottlenecks and challenges of preclinical methods is the first step towards defining a shared PM development strategy, and can lay the foundation for more successful clinical trials across the sector.

The scope of this review is a broad focus on the current preclinical methodologies used to support PM approaches. We aim to highlight the advantages and disadvantages of applying the existing preclinical model systems in the domain of PM and evaluate the status of emerging models. As translational research is a very broad topic, we decided to concentrate on two case models to make the search more manageable. We chose oncology and brain disorders, specifically neuropsychiatric, neurodegenerative, and neurodevelopmental disorders. We reviewed in vivo, in vitro, and in silico methods, and we assessed the models for relevance, validity, and predictive value in the context of PM, highlighting advantages and disadvantages.

This scoping review is part of the PERMIT project (PERsonalized MedIcine Trials), which aims to map the methods for personalized medicine research and build recommendations for robustness and reproducibility of different stages of the development programs. Although several categorization may be proposed, the PERMIT project considers four main building blocks of the personalized medicine research pipeline: (1) design, building and management of stratification and validation cohorts; (2) application of machine learning methods for patient stratification; (3) use of preclinical methods for translational development, including the use of preclinical models used to assign treatments to patient clusters; and (4) evaluation of treatments in randomized clinical trials [22,23,24]. This scoping review covers the third building block in this framework.

## 2. Materials and Methods

We conducted a scoping review, following the methodological framework suggested by the Joanna Briggs Institute [25,26,27]. The framework consists of six stages: (1) identifying the research questions; (2) identifying relevant studies; (3) study selection; (4) charting the data; (5) collating, summarizing, and reporting results; and (6) consultation.

The scoping review approach was considered by the PERMIT consortium to be the most suitable to respond to the broad scope of the field. Compared to systematic reviews that aim to answer specific questions, scoping reviews are used to determine the scope of available evidence in a given field and examine how research is conducted in that field, and they are useful to examine areas that are emerging, to clarify key concepts, and to identify gaps [28]. A study protocol was published in Zenodo before conducting the review (https://zenodo.org/record/3770937, accessed on 30 April 2022) [29]. Due to the iterative nature of scoping reviews, deviations from the protocol are expected and duly reported when occurred. We used the PRISMA-ScR (Preferred Reporting Items for Systematic reviews and Meta-Analyses extension for Scoping Reviews) checklist to report our results [30].

### 2.1. Research Questions

The main research questions addressed were:Which preclinical models are currently used to provide validity data prior to therapeutic clinical trials of PM in oncology and brain disorders and what are the pros and cons of the applied methods?Are the current preclinical models predictive for the outcome of PM trials?

### 2.2. Study Identification

Relevant studies and documents were identified, balancing feasibility with the breadth and comprehensiveness of searches. We searched PubMed, EMBASE, the Cochrane Library, Web of Science, and PsycInfo (search dates: March–June 2020) for research papers and (systematic) reviews in the fields of brain disorders and oncology to identify the most common methodological approaches. Appendix A reports the search strategies applied. We limited our search from 2005 to April 2020 and restricted inclusion to English, French, German, Italian, and Spanish languages. We also searched for gray literature [31] on relevant websites and by consulting partners of the PERMIT project.

### 2.3. Study Selection and Eligibility Criteria

The title and abstracts of records identified by the literature search were screened by two independent reviewers (EO, VF) using the Rayyan online tool [32]. The full-text publications of the relevant articles related to oncology were retrieved and examined by VF, whereas EO retrieved and assessed the articles related to brain disorders to confirm eligibility. Discrepancies were solved by discussion among the review team and the methods group (CG, RB, MF).

Research papers and reviews describing preclinical methods (i.e., cellular, organoid, animal, and in silico models) were considered in the broad context of PM development. We focused on reports assessing the validity, reliability, and predictive value of these methodologies applied to the two case models used in this study, oncology and brain disorders. No restrictions in terms of types of publications were included. References with a focus on disease specific issues, or which did not focus on a personalized approach of the methodology, were excluded. We also excluded congress reports and abstracts.

### 2.4. Charting the Data

We designed a data extraction form using an Excel file (Appendix A). The general study characteristics extracted were as follows: first author name, title of article, year of publication, and type of publication. In addition, for each preclinical model referred to in the paper, we collected information on its definition, the preclinical model type, methodology, advantages, disadvantages, internal and external validity—as previously defined—and applications in PM. During the data extraction phase, the aspects covered by one or more research questions were summarized in tables by one reviewer for each search (VF, EO). Since many narrative reviews have been published about preclinical models, we decided to extract data first from reviews, adding relevant missing information from the remaining research papers. This aspect was not specified in the protocol, but was agreed among the authors before starting the extraction phase process. Assessment of the methodological quality of individual studies included in the analysis was not within the remit of this scoping review.

### 2.5. Consultation Exercise

The members of the PERMIT consortium, associated partners, and the PERMIT project Scientific Advisory Board discussed the preliminary findings of the scoping review in an two-hour long online workshop, held on 2 December 2020.

### 2.6. Patient and Public Involvement

The European Patients’ Forum is a member of the PERMIT project. Although not directly involved in the conduction of the scoping review, they received the draft review protocol to collect comments and feedback.

## 3. Results

### 3.1. Study Selection and General Characteristics of Reports

A total of 1292 records were identified from the oncology search, with an additional 14 records identified through manual search. After removal of duplication, 1158 records remained, and initial screening left 263 articles for full-text evaluation. A final total of 63 studies (3 systematic reviews, 54 narrative reviews, and 6 research papers) met the inclusion criteria and were reviewed for the analysis. For the brain disorders use case, we identified 1516 articles through the literature search and 22 additional records through manual search. After screening of the 1473 unique articles, a full-text review was performed on 263 articles. A total of 94 studies (54 reviews and 40 research papers) met the inclusion criteria and were included in the qualitative synthesis. Figure 2 reports the process for article selection. A list of the included studies can be found in the open access database Zenodo (https://zenodo.org/record/6087847, accessed on 30 April 2022).

The year of publication ranged from 2005 to 2020. The median year was 2017 for oncology, and for brain disorders, the range was 2008 to 2020, and the median was 2016. The types of papers reviewed were mainly narrative reviews (86% in the oncology search and 87% in the brain disorder search); only 5% were systematic reviews in the oncology search, no systematic reviews were identified in the brain disorders search. The remaining papers were research articles.

The literature search, including searching gray literature, did not yield any results in either use case (brain disorders and oncology) where the specific preclinical data that provided the support for approving a clinical trial in PM were reported. Therefore, the included papers were mainly narrative reviews and research articles describing preclinical models and their potential application of PM development. Below, we provide a brief synthesis of the main application towards personalized medicine for the different model systems in two selected use cases. Detailed analysis about specific methods was beyond the scope of this study.

### 3.2. In Vivo Models for PM

Animal models were the most described model for PM approaches, comprising 47% of references in the oncology use case, of which 2/3 related to mouse models. The remainder described various, less commonly used animal-derived models such as chicken chorioallantoic membrane, zebrafish, and companion animal comparative oncology (the study of naturally developing cancers in pets as models for human disease). In the brain disorders use case, 62% of papers referring to animal models were included in the qualitative synthesis. Among them, more than half described rodent models and the rest regarded less commonly used animal models, such as zebrafish, tapeworm, and fruit fly, among others.

Animal models can provide an important instrument for understanding pathogenic mechanisms, identifying drug targets, and developing new therapeutic approaches, but challenges remain for recapitulating the human phenotypes of diseases and to discriminate between successful and unsuccessful treatments [33]. In oncology, the field has progressed some way towards meeting these requirements through the development of patient-derived xenografts (PDXs), where patient tumors are implanted directly into immunodeficient mice [34]. These mouse models largely encapsulate the inter- and intra-tumor heterogeneity observed in cancer, and patient stratification modeling has been attempted in these models; nevertheless, the approach is limited by variable engraftment rates and issues relating to the validity of disease representation in the model [35]. One of the main limitations of PDXs is their immunodeficient status, a prerequisite to facilitate xenotransplantation, but which limits the evaluation of immunological effects. To overcome this, various methodologies are employed to generate a competent human immune system in these models, which are called humanized PDX, leading to different degrees of immune reconstitution. Currently, the major limitation of this approach is the durability and quality of engraftment of the human immune system [11,36,37,38]. Another approach for preclinical modeling of patient stratification in oncology is the co-clinical trials approach, which has been achieved using both PDXs and GEMMs, but the time and cost involved limit the applicability [39].

None of the articles about brain disorders described animal models which can recapitulate the heterogeneity of the human disease phenotypes. Very recent studies in synucleinopathies, diseases characterized by pathological accumulation of aggregated asyn protein, describe animal models that incorporate the two aspects of α-synuclein pathobiology, attempting to reproduce the phenotypic differences seen in the clinical setting. The predictive value of these models remains to be defined, but it can be a first step for the development of preclinical personalized approaches in the field [40,41]. The opportunity to develop chimeric humanized animal models using patient-derived CNS stem cells in vivo has been explored [15], but this technology needs to be further developed and validated before the potential in PM endeavors can be realized.

Advantages and disadvantages of the application of rodent models in PM are summarized in Table 1.

### 3.3. In Vitro Models for PM

Cell culture techniques have been used by researchers for more than 100 years. For much of that time, two-dimensional (2D) monolayer cultures were the gold standard in determining the in vitro efficacy and safety of drug candidates. In the last few years, the emergence of in vitro techniques that can more accurately recapitulate the physiologic features observed in patients, such as three-dimensional (3D) cell culture and organoids, have brought the promise of personalized cellular approaches. Cellular models for PM were included in 18% of the oncology, and 28% of the brain disorders references. In oncology, patient-derived cellular models and tumor explants have been used successfully in drug screening for the most effective therapy [42,43,44]. In brain disorders, the same has been attempted with human lymphoblastoid cell lines (LCL) and induced pluripotent stem cells (iPSC) [16,45,46,47,48].

Organoids can be established to form healthy organs through stem cell initiation, or from tissues directly derived from the patient, and they have the potential to provide disease modeling for infectious disease, genetic disease, PM, drug discovery (screening and toxicology), and regenerative medicine [49,50]. The organoid model is increasingly used for PM in oncology and is described in 23% of the references. Patient-derived organoids have been used as a tool to predict chemotherapy response in individual patients [51,52,53,54]; however, the main disadvantage of this model for the personalized approach is the inconsistency in the organoid growth rate, and the possibility of overgrowth of non-tumor cell populations. In addition, the organoid model does not provide any information about toxicology. Organoid development represents a breakthrough for the study of brain function, evolution, and disorders [55]; however, in our search, only 3% of references referred to brain organoids in PM, and the availability of tissue is a limiting factor [56,57,58].

All these cellular models are limited by the lack of perfusion and biochemical and physical interaction with the surrounding microenvironment. The development of organ-on-chip technologies attempts to overcome some of the limitations of cellular models, and more accurately model personalized drug therapy [13]. Microphysiological systems (MPS), where engineered organ-on-chip technologies are combined with organoids, have the potential to facilitate assessment of pharmacological and toxicological effect [59,60,61]. In oncology, MPS tumor models can replicate the tumor microenvironment in a physiologically relevant manner by incorporating a vascular system, co-culturing with relevant cell types, mimicking elevated interstitial fluid pressure and shear stresses [62]. In brain disorders, the ability to model disease features, microenvironmental parameters, and the complexity of the human central nervous system is highly dependent on the chip [63]. MPS models hold great promise for the future; however, there are still technical, regulatory, and ethical challenges to overcome before patient-derived organ chips are available for clinical evaluation of PM strategies [64]. For a summary of in vitro methods for PM, refer to Table 2.

### 3.4. In Silico Models

In silico modeling, a process of integrating machine learning approaches to biological analysis and simulation, aims to make predictions on drug targets, drug efficacy, and patient responses [20]. This is an emerging field for PM, comprising 12% of the records in oncology, and 8% in brain disorders describing this approach. In silico models in PM aim to couple clinical data with mathematical methods to create subject-specific organ models and design new, personalized protocols, as well as to determine patient stratification. Departing from different patient-specific parameters, they can capture inter- and intra-patient variability [21]. Despite the enormous opportunities offered by these models, a full-scale adaption of patient-specific implementation is still far from reality; the main limitation relates to the prediction accuracy, which depends on the quality and quantity of the input data, and the lack of standards and model validation [65]. The current pros and cons are summarized in Table 3.

### 3.5. Are the Current Preclinical Models Predictive for PM Trials?

In this scoping review, we were not able to identify any articles which directly report on the success rate of PM trials based on accrued preclinical evidence. However, we found several reviews referring to the issues facing preclinical research when applying PM in both oncology (*n* = 6) and brain disorders (*n* = 21). In summary, advancing drug development and biomarker research in the era of PM is highly dependent on choosing the right preclinical model for the right molecular pathway to be explored [66]. Some authors expressed the opinion that retrospective analysis of the preclinical data used to support a failed clinical program should be published to help advance the field [67,68], and others raised the question of whether the more advanced models fit within the established drug-development paradigm, calling for a rethink of the existing anticancer drug discovery pipeline [69].

### 3.6. Main Gaps Identified

To allow for safe development and implementation of PM, appropriate preclinical models generating reliable and predictive data need to be available. Despite the progress in evolving numerous sophisticated model systems, to date, there are fundamental deficits that prevent their broad implementation in PM. As part of the consultation phase of the scoping review, a Gap Analysis Workshop was organized. This workshop took place on 1 December and 2 December 2020 online, and was attended by representatives of all project beneficiaries, as well as by three associated partners, the European Medicines Agency (EMA), the European Network for Health Technology Assessment (EUnetHTA), and the Clinical Trials Coordination Group (CTCG-HMA). From this process, we have identified five main gaps in translational methods, which we believe must be addressed to further develop robust models for PM.

The first gap is a lack of clinically relevant experimental models for personalized medicine. Despite technical advances and more sophisticated preclinical models, to date, there are knowledge gaps in biology and an inability to recapitulate human phenotypes for many diseases, which is a challenge for translation and prediction of preclinical data to human PM clinical trials. There is also an apparent deficit in validating preclinical methods for clinical relevance; in other words, how well the model represents the phenotype of disease or clustering of patients.The second gap is the lack of standards for methods, validation procedures, and the lack of quality assessment systems. The fact is that preclinical models are often not robust enough for translation. Some of the hurdles for model validation are that this type of work is not academically rewarded, it is time consuming, and it is expensive.The third gap is the lack of accurate reporting and the lack of reporting negative results, which then further leads to a lack of systematic reviews and meta-analyses on methods, and these are important tools for evidence-based medicine. Access to preclinical data supporting clinical trials is challenging. There is a publication bias toward positive experiments, and methods are often not reported in sufficient detail to attempt reproducibility of results.The fourth gap relates to regulation, and the lack of harmonized guidelines for evaluating the relevance and robustness of preclinical evidence.The last gap we identified is the lack of involvement between preclinical and clinical research, and the need for a better definition for patient engagement.

## 4. Discussion

### 4.1. Principal Findings

Traditionally, preclinical models have been used as simplified models of human conditions, managed with a high degree of standardization, kept in pathogen-free environments, and treated identically to remove the influences of known variables. The increasing multi-omics characterization of disease, generated by advances in molecular characterization and bioengineering, brings new challenges to disease modeling, which becomes even more evident when associated with personalized treatment decisions. The complexity of PM highlights the need for more sophisticated biological systems to assess the integrated mechanisms of response. This scoping review investigated how current preclinical methods can support decision makers in approving clinical trials in PM. In the field of oncology, where the personalized approach is the most advanced, preclinical models which can recapitulate the patient tumor heterogeneity exist; nevertheless, the approach of modeling patient clustering through this approach is not yet widely used for various reasons. In brain disorders, there is no availability of models which can fully recapitulate patient phenotypes, and there is a dearth in the understanding of the disease mechanisms occurring at an individual level. Emerging models, such as organ-on-chip and in silico models, have been proposed to close the translational gap in the future. However, this is reliant on technologies which are still in their infancy, and additional fundamental issues in preclinical research remain unsolved.

### 4.2. Limitations of the Scope

As the preclinical research topic is broad, we decided to concentrate on two case studies to make the search manageable. These were oncology and brain disorders. It might have been informative to also include other disease areas, but we decided to narrow our focus to the two use cases, which possibly represent the two extremes in relation to availability of preclinical models in PM. The main limitation of our scope is that we were unable to find information about the specific preclinical evidence supporting the decision to approve clinical trials, and therefore we could not directly assess the translatability of the model used. One of the reasons for this is probably that most clinical trials are sponsored by industry, and preclinical data generated by the pharmaceutical industry are often not published. Through our gray literature search, we could identify two registries for preclinical trial protocols (www.preclinicaltrials.eu, accessed on 6 July 2022; www.animalstudyregistry.org, accessed on 6 July 2022), but there is no requirement to use such registries. The number of registered preclinical studies is 118 and 113, respectively, at the time of publication, and they are not linked to subsequent clinical data. In contrast, there is a legal requirement to register clinical trials; a search on clinicaltrials.gov at the same timepoint showed 420,268 entries. This scoping review was primarily aimed to inform the development of the subsequent recommendations, and even if the search might be perceived as out of date, we have continued to monitor the literature in the field during these two years and added relevant papers.

### 4.3. Challenges of Preclinical Research in PM

The probable gaps identified in this review are not novel issues in preclinical research [70,71]. The low rate of translation is evident when looking at the high attrition rates in drug and medical device development, and it has been suggested that this could be explained by faults in the design, conduct, and reporting of preclinical studies [7,72,73,74,75]. Attempts at addressing the issues relating to preclinical study reporting have been made through the development of reporting checklists, such as the ARRIVE guidelines for in vivo experiments [76,77], but despite being endorsed by over 1000 scientific journals, non-compliance with standard reporting checklists was the major finding in three systematic reviews of PDX models, where only one study was found to fully comply with the guidelines [36,78,79].

Furthermore, the systematic validation of the model systems often fails, in terms of internal, external, construct, and predictive validity [80]. The internal validity in preclinical methods, i.e., the risk of bias, can be addressed by systematic reviews, which can improve the success and reproducibility of subsequent translational clinical studies in this era of PM. However, only three systematic reviews were identified in the oncology search and none in the brain disorders search, and this deficit in preclinical research has already been addressed by the SYRCLE initiative [81]. The absence of standardized protocols and guidelines, because of the large variation in methodology across preclinical studies, makes quantitative analysis of bias challenging across studies, and researchers have made calls for more harmonized approaches [82]. Another relevant point to be addressed is the low availability of negative data. Negative results are not appealing for publication, meaning that the results of thousands of experiments that fail to confirm the reliability of preclinical models do not see the light of day. However, this is not necessarily a result of publication bias with the journals, but rather that scientists do not submit negative studies [83,84,85]. It results in a waste of time and resources from other scientists in repeating negative findings and, consequently, the deceleration of the translational pipeline. This is even more true in an industry setting, where in-house data are not generally published for reasons of competitiveness [86,87]. Therefore, the scientific community should address this issue, showing awareness of the richness of negative results in research.

The predictive validity of preclinical research is challenging when modeling complex disease processes. One approach could be to test the hypotheses in several different models, which could capture various aspects of the heterogeneity of the human pathophysiological processes. Another is to make sure that the model adequately represents the human disease condition for the question being asked. Several tools and guidelines for assessing the clinical relevance preclinical research have been published [88,89,90,91], but so far, they have not been broadly implemented. In the end, it is important to remember that a model system can never be a complete reflection of a human. However, by choosing the most appropriate model for the question asked, striving to make sure that the model is fully validated, and using complementary models to fill in the gaps, enough evidence can be gathered to move through the translational phase. To an extent, it is the drive to develop animal-free methods which is fueling the development of advanced in vitro and in silico models. Animal research is strictly regulated from an ethical point of view, but not from a qualitative perspective, and this deficit is becoming more evident when in vivo models are applied for PM [92].

The disappointing results of PM clinical trials cannot entirely be attributed to the lack of preclinical models that recapitulate human disease phenotypes. Continued efforts should be directed towards deep phenotyping of patients, and identification of reliable biomarkers to identify patient subgroups [70], as well as better definition of criteria for patient selection in clinical trials [93].

Based on the findings of this scoping review, it is our opinion that all preclinical research will benefit from better guidance regarding the clinical relevance and validity of models. When preclinical methods are applied to personalized medicine development, such as modeling molecular analyses of patient samples and treatment outcomes, the gaps become even more evident. These challenges need to be addressed at global, national, regional, and local levels.

## 5. Conclusions

When adequately designed and conducted, preclinical experiments may contribute invaluable information to our knowledge of medicine, including the discovery and development of new drugs, and can be essential tools to bridge the translational gap between preclinical and clinical research. In fact, appropriate preclinical models should be an integral contributor to interventional clinical trial success rates. We are at a key moment for the era of PM, and research in this domain is constantly evolving and generating new knowledge. However, PM development has proven to be an extremely ambitious enterprise at the preclinical level. The gaps identified in this scoping review are the first step towards building recommendations for more robust translational research in PM and corresponding to different scientific domains. The challenges can only be faced with concomitant action on all levels, with the involvement of all relevant stakeholders involved in this field, from researchers to policy makers and regulators, and should be viewed as aspects to work on, rather than obstacles, as they form the foundation of personalized preclinical research.

## Figures and Tables

**Figure 1 jpm-12-01177-f001:**
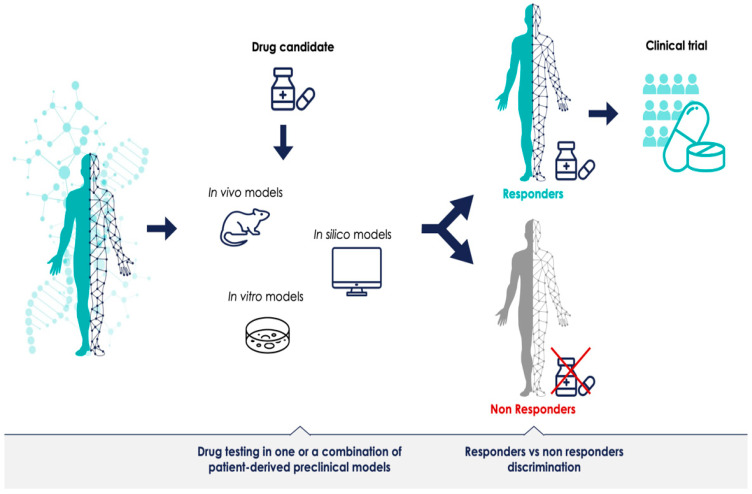
Predictive patient-derived translational models for personalized medicine. Preclinical development in clinically relevant models with robust predictions could improve clinical trials for personalized medicine.

**Figure 2 jpm-12-01177-f002:**
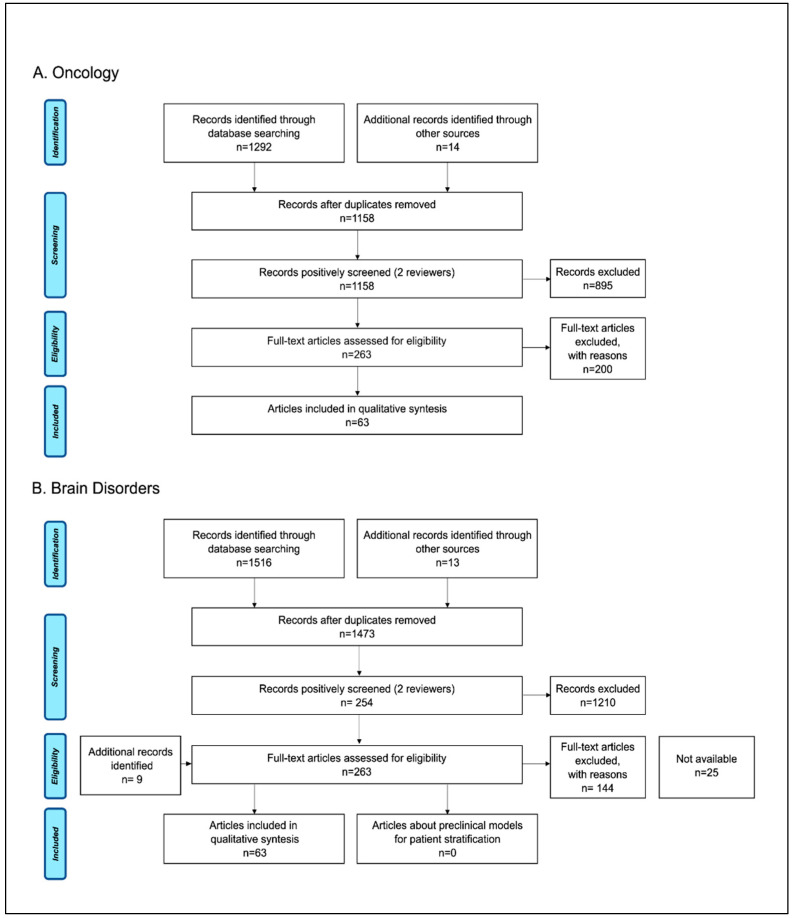
Study selection flow diagram: PRISMA flow-chart of data collection process for the (**A**) oncology and (**B**) brain disorders literature searches.

**Table 1 jpm-12-01177-t001:** Rodent models for personalized medicine.

Use Case	Advantages	Disadvantages
**Oncology**	**PDX ^1^** Recapitulate the intra- and inter-tumor heterogeneity of human cancer Suitable for biomarker discovery Can perform personalized drug screening of individual patient tumors Can establish large collaborative PDX platforms	**PDX ^1^** Engraftment-induced molecular divergence from the original tumor Cannot study metastases in subcutaneous models Lack of stromal and immune compartments Variable and unpredictable engraftment rates High cost, technically challenging Lack of standardized protocols
	**GEMM ^2^** Allow de novo tumor formation that recapitulates molecular and histopathological features of human disease in a native immune-proficient microenvironment	**GEMM ^2^** Very long development time; transition of data to the clinic is slow Mouse tumor with reduced clonal heterogeneity compared with human tumors
**Brain disorders**	Suitable for studying the side-effects of chronic drug administrations Suitable for biomarker discovery Suitable for chimeric models Development of behavioral assays that can be used to evaluate drug efficacy at the behavioral level	Lack of biological understanding Do not fully recapitulate the pathophysiology of human conditions and human phenotypes Use of young animals is not representative for pathology Species-to-species differences Limited in predicting treatment efficacy in human disorders

^1^ PDX—patient-derived xenograft; ^2^ GEMM—genetically modified mouse models.

**Table 2 jpm-12-01177-t002:** In vitro methods for personalized medicine.

Use Case	Advantages	Disadvantages
**Oncology**	**2D monolayer cell culture** Easy production for high throughput screening procedures	**2D monolayer cell culture** Static model Do not represent heterogeneity of tumor tissues.
	**3D tumor cultures** Recapitulate the in vivo tumor architecture more closely than 2D models, including cell morphology, growth kinetics, signaling pathways, and drug response	**3D tumor cultures** Technically challenging; inconsistent growth rates. Not able to capture the intra-tumor heterogeneity of patient samples Lack of standardized protocols
	**Organoids** Can be generated from individual cancer patients Patient-derived organoids are cellularly and molecularly representative of parent tumor Can perform drug screening of individual tumors Can establish biobanks of organoids for drug discovery Can be transplanted for in vivo screening Less expensive than PDX models	**Organoids** Inconsistent growth rate, overgrowth of normal epithelial cells Lack of stromal and immune compartments, lack of perfusion, lack of tumor micro-environment Direct drug exposure of tumor, not representative for humans Low throughput, medium requirements limiting factor Not validated to replace existing systems Technically challenging, expensive, access to tumor material
	**Organ-on-chips** Potential to facilitate assessment of pharmacological and toxicological effects Replicate the tumor microenvironment in a physiologically relevant manner	**Organ-on-chips** Technical and ethical challenges Lack of standardized protocols
**Brain disorders**	**LCL ^1^** High value for drug response biomarker discovery Available in large public biobanks Low cost	**LCL ^1^** Do not reflect the in vivo conditions
	**iPSC ^2^** Patient genetic background, overcomes inter-species differences Can be differentiated into different CNS cell types Good platform for high-throughput screening for drugs and for toxicology tests Suitable for chimeric humanized animal models	**iPSC ^2^** High cost, labor intensive Variable reproducibility Lack of standardized and validated protocols Do not recapitulate the brain architecture
	**Organoids** Dimensional complexity Model the 3D structure, organization, composition, and connectivity of the human brain. Resemble the early developing human brain with respect to gene-expression programs Exhibit human-specific cellular diversity, histological layers, and migration patterns	**Organoids** High variability in growth rates Do not recapitulate the precise organization of the brain Lack of maturity and limitations in the cellular composition. High costs Lack of validated protocols Lack of reproducibility

^1^ LCL—human lymphoblastoid cell lines; ^2^ iPSC—human-induced pluripotent stem cells

**Table 3 jpm-12-01177-t003:** In silico models for personalized medicine.

Use Case	Advantages	Disadvantages
**Oncology**	Prediction of drug effects and functional responses based on mathematical methods Possibility of refining experimental programs of clinical and biomedical studies involving laboratory work, resulting in a reduction in animal experiments	Unknown parameters affect accuracy of prediction Lack of standards for data quality and methodology
**Brain disorders**	Suitable for coupling clinical data with mathematical methods to create subject-specific brain models to design new, personalized, and more optimal protocols Departing from patient-specific parameters ability to capture inter- and intra-patient variability, the difference between patients, and the evolution of patient condition	Inadequacy of key sensitivity parameters Lack of guidelines for obtaining high-quality data Lack of model validation

## Data Availability

Copies of searches and data extraction sheets are publicly available on the online platform Zenodo (https://zenodo.org/record/6087847, published on 15 February 2022), as part of the database collection for all scoping reviews conducted in the PERMIT project.

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
