# Peer review of "Evaluating Translational Methods for Personalized Medicine—A Scoping Review"

_jpm, 2022, doi:10.3390/jpm12071177_

Round 1

Reviewer 1 Report

The authors have performed a rigorous scoping review on personalized medicine for two case studies. In my opinion, the lack of preclinical data supporting clinical trials is a bit of an “inconvenient truth” for the fields of laboratory animal and clinical sciences, which should be communicated widely. The paper reads well, and methodologically it is one of the best I’ve been asked to peer-review. I happily recommend it for publication after minor revisions.

General:

The authors’ work definition of “personalised medicine” is clear. To increase overall understanding of the paper, please add definitions of “translation” and “predictive validity”.

The authors seem convinced of the value of animal studies throughout. Many scientists within the field agree with them, but after e.g. doi: 10.1186/s12967-019-1976-2 combined with a number of papers showing higher predictive validity of in vitro than in vivo models, I am no longer too convinced. The phrasing could be nuanced a bit throughout, but particularly L224-225 has a firm statement on the value of animal studies which needs either a reference or “hedging”. 

Minor comments:

P3 L120-121: This phrasing sounds more compatible with the definition of a mapping review. Please adapt.

L199 section 3.1: Please extend the sample description. You extracted data on many more paper- and study characteristics that would be informative to show somehow; categorical variables can be summarized in percentages and maybe also add range and- or median year of publication.

Table 2 disadvantages of 2D models: The “lack of predictive value” implies that the other methods do have predictive value. If so, this needs referencing and a definition (definition also requested above).

L391: were these study registries checked for ongoing and planned relevant studies? It would be great to get an idea of the overall size of the field (also for clinical registries).

L399: With a scoping review, literature might have been missed. Please add “probable” in front of gaps.

L419: Indeed, and yet scientists often blame the journals for publication bias. It might be nice to refer to literature showing that the journals are not at fault (DOIs: 10.1001/jama.287.21.2825   10.5694/j.1326-5377.2006.tb00418.x    10.2106/JBJS.G.00279).

L423-424: may sound likely, but this statement needs a reference.

L451-454: Here again some hedging is needed. For the first part, simply replacing “can” by “may” or “might” could work.

Search: the search is well-described, but why was Embase not searched for “:ti,ab,kw”? This would have been symmetrical with “[tiab]” in PubMed, which gives you the author-defined keywords for free.

Author Response

We would like to thank the reviewer for the kind comments and for the constructive feedback. We have tried to take all the specific comments into consideration for improvement of the manuscript.

The authors’ work definition of “personalised medicine” is clear. To increase overall understanding of the paper, please add definitions of “translation” and “predictive validity”.

Thank you for pointing out this oversight on our behalf. We have added a definition of what is meant by “translational research” on L 47-47, and a definition for “predictive value” on L 84.

The authors seem convinced of the value of animal studies throughout. Many scientists within the field agree with them, but after e.g. doi: 10.1186/s12967-019-1976-2 combined with a number of papers showing higher predictive validity of in vitro than in vivo models, I am no longer too convinced. The phrasing could be nuanced a bit throughout, but particularly L224-225 has a firm statement on the value of animal studies which needs either a reference or “hedging”. 

Thank you for this comment. We agree with the reviewer that inappropriate animal models contribute to high attrition rates in the development of human therapies, and we have added the word can into the mentioned sentence (now L 235-235) and added this reference; doi: 10.1186/s12967-019-1976-2.

P3 L120-121: This phrasing sounds more compatible with the definition of a mapping review. Please adapt.

The sentence has been changed to: “scoping reviews are used to determine the scope of available evidence in a given field, examine how research is conducted in that field, and they are useful to examine areas that are emerging, to clarify key concepts and identify gaps” L123-125, according to reference doi: 10.1186/s12874-018-0611-x.

L199 section 3.1: Please extend the sample description. You extracted data on many more paper- and study characteristics that would be informative to show somehow; categorical variables can be summarized in percentages and maybe also add range and- or median year of publication.

The following paragraph has been added in the results section 3.1, L 202-207: The year of publication ranged from 2005 to 2020, with the median year 2017 for oncology, and for brain disorders the range was 2008 to 2020, and the median was 2016. The types of papers reviewed were mainly narrative reviews (86% in the oncology search, and 87% in the brain disorder), only 5% were systematic reviews in the oncology search, no systematic reviews were identified in the brain disorders search. The remaining papers where research articles.

Table 2 disadvantages of 2D models: The “lack of predictive value” implies that the other methods do have predictive value. If so, this needs referencing and a definition (definition also requested above).

Thank you for pointing this out. This has now been changed in Table 2 to “Do not represent heterogeneity of tumor tissues”.

L391: were these study registries checked for ongoing and planned relevant studies? It would be great to get an idea of the overall size of the field (also for clinical registries).

We have checked the registries and added the following: “The number of registered preclinical studies is 118 and 113, respectively, at the time of publication, and they are not linked to subsequent clinical data. In contrast, there is a legal requirement to register clinical trials, a search on clinicaltrials.gov at the same timepoint showed 420 268 entries.” L 407-410.

L399: With a scoping review, literature might have been missed. Please add “probable” in front of gaps.

Done. Now L 416.

L419: Indeed, and yet scientists often blame the journals for publication bias. It might be nice to refer to literature showing that the journals are not at fault.

Thank you, good point! We have added the following sentence on L 438-439: “However, this is not necessarily a result of publication bias with the journals, but rather that scientist do not submit negative studies”  (Ref. doi: 10.1001/jama.287.21.2825, 10.5694/j.1326-5377.2006.tb00418.x, 10.2106/JBJS.G.00279)

L423-424: may sound likely, but this statement needs a reference.

Thank you for pointing out this oversight. We have added these references, doi: 10.1002/ijc.32405, 10.1111/bph.12771, L 443.

L451-454: Here again some hedging is needed. For the first part, simply replacing “can” by “may” or “might” could work.

We have amended the sentence accordingly, L 474-477: “When adequately designed and conducted, preclinical experiments may contribute invaluable information to our knowledge of medicine, including the discovery and development of new drugs, and can be essential tools to bridge the translational gap between preclinical and clinical research. In fact, appropriate preclinical models should be an integral contributor to interventional clinical trial success rates.”

Search: the search is well-described, but why was Embase not searched for “:ti,ab,kw”? This would have been symmetrical with “[tiab]” in PubMed, which gives you the author-defined keywords for free.

We tried the combination Ti/Ab/Kw on Embase but the number of records increased very much, probably because author-selected keywords often are very generic. Keywords generally reflect concepts included in the abstract, thus we believe the impact of not searching that specific field on search sensitivity is likely to be negligible. 

Reviewer 2 Report

This review on the importance of comprehensive disease-specific translational models for validation of disease-specific treatment (also called personalized medicine) is highly relevant. This is especially important in the field of neurodegenerative disease where patients exhibit heterogeneous and overlapping clinical phenotypes, limiting accurate and early diagnosis. As the authors highlight, treatment validation using a one-model-fits-all approach, may explain why successful treatment validation studies in animal models, continue to yield disappointing results in clinical trials. 

I have a few minor comments:

*Line 243: ''None of the articles about brain disorders described animal models which can recapitulate the heterogeneity of the human disease phenotypes.'' --> Recently, more and more studies aim to recapitulate body-first and brain-first Lewy body disorders with focus on both CNS and PNS (reviewed in PMID: 34952161, PMID: 35693346).

*Table 1: Disadvantages of in vivo models of brain disorders: I would add the consistent use of using young rodents to investigate disease mechanisms in brain disorders where age is the largest risk factor. It has been shown that aged rodents are more susceptibility to develop pathology and neurodegeneration, and present with a more accelerated disease course, compared to young animals (PMID: 33880502). Consequently, studies in young animals may not be an accurate reflection of human neurodegenerative diseases, limiting translational outcomes. Importantly, the few treatment validation studies in rodent models of neurodegenerative disease that directly compare young and old rodents demonstrate reduced efficacy in older rodents. For example, in cell transplant treatment studies, a decreased survival of grafted dopaminergic neurons in aged, compared to young neurotoxin-lesioned rats has been observed.

*The disappointing results of PM in clinical trials cannot entirely be attributed to the lack of preclinical models that recapitulate complete human disease phenotypes. Patients with neurodegenerative disease are often misdiagnosed, and are diagnosed at more advanced disease stages, characterized by major neurodegeneration. Clinical trials often include patients with advanced neurodegeneration and from different neurodegenerative disease entities due to misdiagnosis. It is conceivable that PM or disease-specific disease-modifying treatment will appear less effective in a patient group that is characterized by major neurodegeneration and entails different neuropathological and clinical profiles. Consequently, poor patient selection may also contribute to disappointing results in clinical trials (PMID: 35204668). Further research towards biomarker development for early disease stratification will enable proper patient selection and identification of disease-specific treatment targets. Such research would also require the development of animal models that recapitulate the different human disease phenotypes (and not 'one-model-fits-all' approach). I believe a short paragraph on this topic would benefit this review.

Author Response

We would like to thank the reviewer for the kind comments, and the suggestions for improvements, which we have tried to implement.

*Line 243: ''None of the articles about brain disorders described animal models which can recapitulate the heterogeneity of the human disease phenotypes.'' --> Recently, more and more studies aim to recapitulate body-first and brain-first Lewy body disorders with focus on both CNS and PNS (reviewed in PMID: 34952161, PMID: 35693346).

Thank you for bringing these recent reports to our attention. The search was performed in April 2020, which explains why we did not include them. We have added the following paragraph on L 254-259, based on the references you supplied: “Very recent studies in synucleinopathies, diseases characterized by pathological accumulation of aggregated asyn protein, describe animal models that incorporate the two aspects of α-synuclein pathobiology, attempting to reproduce the phenotypic differences seen in the clinical setting. The predictive value of these models remains to be defined, but it can be a first step for the development of preclinical personalized approaches in the field”

Table 1: Disadvantages of in vivo models of brain disorders: I would add the consistent use of using young rodents to investigate disease mechanisms in brain disorders where age is the largest risk factor. It has been shown that aged rodents are more susceptibility to develop pathology and neurodegeneration, and present with a more accelerated disease course, compared to young animals (PMID: 33880502). Consequently, studies in young animals may not be an accurate reflection of human neurodegenerative diseases, limiting translational outcomes. Importantly, the few treatment validation studies in rodent models of neurodegenerative disease that directly compare young and old rodents demonstrate reduced efficacy in older rodents. For example, in cell transplant treatment studies, a decreased survival of grafted dopaminergic neurons in aged, compared to young neurotoxin-lesioned rats has been observed.

Thank you for raising this pertinent issue. We have added “Use of young animals not representative for pathology” under disadvantages in Table 1: Rodent models for personalized medicine.

*The disappointing results of PM in clinical trials cannot entirely be attributed to the lack of preclinical models that recapitulate complete human disease phenotypes. Patients with neurodegenerative disease are often misdiagnosed, and are diagnosed at more advanced disease stages, characterized by major neurodegeneration. Clinical trials often include patients with advanced neurodegeneration and from different neurodegenerative disease entities due to misdiagnosis. It is conceivable that PM or disease-specific disease-modifying treatment will appear less effective in a patient group that is characterized by major neurodegeneration and entails different neuropathological and clinical profiles. Consequently, poor patient selection may also contribute to disappointing results in clinical trials (PMID: 35204668). Further research towards biomarker development for early disease stratification will enable proper patient selection and identification of disease-specific treatment targets. Such research would also require the development of animal models that recapitulate the different human disease phenotypes (and not 'one-model-fits-all' approach). I believe a short paragraph on this topic would benefit this review

Thank you, this is an important point, and we have added the following paragraph on L 461-464: “The disappointing results of PM clinical trials cannot entirely be attributed to the lack of preclinical models that recapitulate human disease phenotypes. Continued efforts should be directed towards deep phenotyping of patients, and identification of reliable biomarkers to identify patient subgroups (doi: 10.1186/s41231-019-0050-7), as well as better definition of criteria for patient selection in clinical trials (doi: 10.3390/biom12020168).”